# Interventional Heartworm Extraction in Two Dogs: The Clinical Application of Impedance Cardiography

**DOI:** 10.3390/ani13193127

**Published:** 2023-10-07

**Authors:** Daesik Kim, Seol-Gi Park, Minsu Kim

**Affiliations:** 1Veterinary Emergency Medicine, Department of Veterinary Clinical Science, Research Institute for Veterinary Science, College of Veterinary Medicine, Seoul National University, Seoul 08826, Republic of Korea; daxic88@snu.ac.kr; 2Incheon Sky Animal Hospital, Incheon 21555, Republic of Korea

**Keywords:** heartworm, cardiac output, non-invasive monitoring, dog, interventional extraction

## Abstract

**Simple Summary:**

The study explored the use of impedance cardiography (ICG) for monitoring cardiac performance during heartworm extraction surgeries in dogs. In two observed cases, ICG effectively captured the heart’s functional shifts before and after the procedure. This non-invasive method may be of potential value for monitoring heart function during such surgeries.

**Abstract:**

Heartworm (HW) disease, caused by *Dirofilaria immitis*, is a life-threatening ailment in dogs. HW disrupts blood flow and decreases cardiac output (CO). The accurate monitoring of CO during HW extraction is pivotal for patient survival and overall health. Objective: This study aimed to assess the efficacy of using impedance cardiography (ICG) as a non-invasive approach for monitoring CO during interventional HW extraction. Methods: Two cases of HW infections were treated via surgical extraction. The CO and mean arterial pressure (MAP) were monitored using the ICG technique during the anesthesia stabilization, extraction process, and post-extraction phases. Results: In Case 1, the CO increased by 115% post-procedure, and in Case 2, the CO increased by 116%. In contrast, the MAP varied between the two cases. The ICG method provided real-time CO data without major disruptions during the extraction surgery. Conclusion: The ICG technique for CO monitoring during interventional HW extractions is effective.

## 1. Introduction

In dogs, heartworm (HW) disease is a parasitic infection caused by *Dirofilaria immitis*. Transmission occurs via mosquitoes. The larvae circulate in the bloodstream and mature over time. As the number of adult HWs escalates, they predominantly inhabit the main pulmonary arteries, right atrium, right ventricle, and the vena cava, potentially precipitating caval syndrome. The presence of these accumulated adult HWs can result in significant hemodynamic alterations, inducing pulmonary hypertension, and reducing cardiac output (CO) [1]. In valvular diseases, retrograde blood flow occurs upon cardiac ejection, leading to a reduced CO [2]. In contrast, a decreased CO in HW diseases is caused by the presence of HW within the heart, obstructing the normal blood flow [3]. These conditions may ultimately lead to secondary congestive heart failure, which is lethal to the host [4,5]. In caval syndrome, pharmacological treatment may trigger unwanted immune responses and blood clot formation as a result of deceased HWs [6]. As a result, surgical extraction is preferred for managing severe HW disease [4,7,8]. As HW directly affects the cardiovascular system, patients are typically classified as ‘at-risk’ for anesthesia [9]. The monitoring of the cardiovascular system is essential for safe surgery and anesthesia [7]. However, despite the anticipated CO changes, the CO during and after extraction surgery has not been investigated to date. In this case report, we described the use of a non-invasive monitoring system to measure the CO changes following HW extraction surgery.

## 2. Case Presentation

### 2.1. Interventional Extraction of HW

To reduce procedural side effects, each dog was medicated, starting from 3 days prior to the procedure and continuing up to the day of the procedure. The medications administered included prednisolone (0.5 mg/kg PO BID; Solondo Tab., Yuhan Co., Cheongju, Republic of Korea), doxycycline monohydrate (5 mg/kg PO BID: Doxycycline Tab., Youngpoong Co., Incheon, Republic of Korea), silymarin (20 mg/kg PO SID; Silymarin Tab., Sinilpharm Co., Seoul, Republic of Korea), famotidine (0.5 mg/kg PO BID; Famotidine Tab., Hanallbiopharma Co., Seoul, Republic of Korea), clopidogrel hydrogen sulfate (2 mg/kg PO SID; Pravic Tab., SinilPharm Co., Seoul, Republic of Korea), cefaclor (20 mg/kg PO BID; Cefaclor Cap., Ilhwa Co., Gyeonggi, Republic of Korea), and furosemide (1 mg/kg PO SID; Lasix Tab., Handok Co., Seoul, Republic of Korea). Pre-oxygenation was performed prior to general anesthesia. For general anesthetic induction, we used alfaxalone (1–4 mg/kg IV titrated to effect; Alfaxan., Careside Co., Gyeonggi, Republic of Korea). Following tracheal intubation, anesthesia was maintained using isoflurane at a minimal concentration, ranging from 1 to 4%. During anesthesia, blood pressure was measured using a veterinary-specific oscillometric non-invasive blood pressure device (Vet20 SunTech; SunTech Medical Inc., Morrisville, NC, USA).

Right jugular venotomy was performed using the HW basket device (S & G Biotech Inc., Seoul, Republic of Korea) for extraction. For the right jugular venotomy, a local block was administered at the site using lidocaine (1 mg/kg SCl; Lidocaine 2%., Huons Co., Gyeonggi, Republic of Korea). The skin near the jugular vein was incised to expose the jugular vein. After securing the vein with sutures, a small incision was made, and the HW basket device was inserted to remove the HWs. The HWs were removed under fluoroscopic guidance. The procedure was continued until no more HWs could be retrieved in two consecutive attempts and an increase in CO was observed.

### 2.2. Impedance Cardiography (ICG)

To measure the ICG, 6 electrodes were placed. Two electrodes were attached to the 4th and 6th cervical vertebrae located dorsally from the left jugular vein of the neck (Z1, Z2). One electrode was positioned to the right of the sternum and connected to the fifth intercostal space (EKG1). Another was attached near the left 13th rib (EKG2). Furthermore, two other electrodes (Z3 and Z4) were affixed just below the xiphoid protrusion (Figure 1). The electrodes were then connected to the PhysioFlow^®^ acquisition system (version 1.0.7 RC9.15 SVV Edition Lab1 Enduro, Manatec Biomedical, Bristol, PA, USA). During the procedure, the stroke volume and CO values were continuously measured to obtain PhysioFlow^®^ files. According to the manufacturer’s instructions, a signal quality of ≥75% of PhysioFlow^®^ is considered to be reliable for clinical use [10]. Any instances of a poor signal quality due to the motion or movement of the basket during the procedure were excluded. Only readings with a signal quality stability of 80% or higher, as indicated by the software, were recorded. The PhysioFlow^®^ measured the CO every 5 s. To minimize the measurement error, the CO and MAP were measured 15 times each during the pre-extraction, extraction, and post-extraction periods, resulting in a total of 45 sets of data.

### 2.3. Case 1

A 5-year-old, intact, female mixed-breed dog, weighing 5.2 kg, was referred with symptoms of anorexia and tachypnea. Clinical signs, such as exercise intolerance, mild dehydration, and a pale mucous membrane were observed. An immunological antigen test was positive for adult worms (Rapid CHW Ag 2.0 kit Bionote Co., Gyeonggi, Republic of Korea) and microfilariae were observed in the peripheral blood smear examination. Thoracic radiography revealed a reverse D heart sign, a vertebral heart size (VHS) of 11.5, and enlargement of the pulmonary artery. Echocardiograms revealed double-walled linear echoic structures, indicating adult HWs in the right heart (Figure 2A). Hematology and serum chemistry revealed hemolytic anemia, leukocytosis, and mildly increased hepatic enzyme levels. ALP 365 U/L (alkaline phosphatase, ref. 18–101), ALT 225 (alanine transaminase, ref. 12–101), and AST (aspartate aminotransferase, ref. 17–44). After visiting the clinic, the patient was given medication for three days to reduce potential side effects before undergoing the HW removal procedure. Following the interventional extraction procedure previously described, the patient exhibited an improvement in clinical symptoms. Seven days after the intervention, an improvement in clinical symptoms (vital signs and exercise intolerance) was observed. Adult HWs were not visualized on the echocardiograms (Figure 2B). The case included monitoring for pulmonary hypertension before and after the procedure using PV Vmax, TR Vmax, and TR max PG values. Before the procedure, the PV Vmax was 1.85 m/s, the PV max PG was 13.73 mmHg, the TR Vmax was 3.91 m/s, the TR max PG was 59.77 mmHg, the E peak velocity was 0.67 m/s, and the E/A Ratio was 1.89, and after the procedure, the PV Vmax was 0.81 m/s. The PV max PG was 2.64 mmHg, the TR Vmax was 2.71 m/s, the TR max PG was 29.29 mmHg, the E peak velocity was 0.73 m/s, and the E/A Ratio was 1.58. Pulmonary hypertension improved from a moderate to severe stage to a mild stage, while it was difficult to conclude that there had been an improvement in the left ventricle diastolic function.

### 2.4. Case 2

A 13-year-old, spayed, female Yorkshire terrier, weighing 3.72 kg, was referred with symptoms of abdominal distension, anorexia, and tachypnea. In radiography, there was evidence of an enlargement of the left and right heart chambers, a reverse D heart shape (VHS 10.4), and a decreased visibility of the diaphragmatic line in the upper abdomen. The echocardiogram revealed HWs inside the right atrium, and severe tricuspid regurgitation was confirmed (Figure 3A). Mitral valve lesions were not identified. Moreover, color-flow Doppler imaging showed no evidence of mitral regurgitation. An immunological antigen test was positive for adult worms (Rapid CHW Ag 2.0 kit Bionote Co., Gyeonggi, Republic of Korea) and microfilaria was observed in the peripheral blood smear examination. The result of the abdominal fluid analysis showed a modified transudate. Seven days after the extraction, no further abdominal fluid was observed. Serologically, the liver enzymes were not elevated. However, there was an increase in WBC (18.7 × 10^3^ cells/µL with a reference range of 6.0–17.0) and a slight azotemia (blood urea nitrogen, 43.6 mg/dL, with a ref. 9.2–29.2) was observed. To manage the right atrial dilation and improve the cardiac function, furosemide (1 mg/kg orally once daily; Lasix tablets, Handok Co., Seoul, Republic of Korea) and enalapril (0.5 mg/kg orally; Enaprin tablets, Jonggeundang Co., Seoul, Republic of Korea) were administered. After visiting the clinic, the patient took prescribed medication for three days to mitigate potential side effects before having the HW extraction. Subsequently, the HW extraction was performed as per protocol. Seven days after the intervention, adult HWs were not visualized on the echocardiograms (Figure 3B). The case included monitoring for pulmonary hypertension before and after the procedure using PV Vmax, TR Vmax, and TR max PG values. Before the procedure, the PV Vmax was 0.79 m/s, the PV max PG was 2.48 mmHg, the TR Vmax was 5.43 m/s, the TR max PG was 117.91 mmHg, the E peak velocity was 0.47 m/s, and the E/A Ratio was 0.78, and after the procedure, the PV Vmax was 0.68 m/s. The PV max PG was 1.87 mmHg, the TR Vmax was 4.16 m/s, the TR max PG was 69.25 mmHg, the E peak velocity was 0.73 m/s, and the E/A Ratio was 0.78. Pulmonary hypertension was still in the severe stage, but an improvement in the speed of TR Vmax was observed.

## 3. Results

In Case 1, a total of 11 HWs were removed through 10 extraction procedures. In Case 2, a total of 16 HWs were extracted during 9 procedures. Both dogs underwent the procedure without complications, and recovered from anesthesia uneventfully. Three days post-procedure, an echocardiogram revealed the absence of HWs in the right atrium and ventricle. Subsequently, the patients received three doses of melarsomine dihydrochloride (2.5 mg/kg IM; Immiticide Inj., Merial Inc., Nanchang, China) during the following 2-month follow-up period. A total of 45 data sets, measuring the CO and MAP 15 times at each time point (anesthesia stabilization, during extraction, and after extraction), were acquired (Table 1).

In Case 1, the average CO during the pre-procedural anesthesia stabilization phase was 0.88 L/min. Post-extraction, the average CO was 1.01 L/min, indicating a 115% increase. Similarly, in Case 2, the average CO in the pre-procedural anesthesia stabilization phase was 0.59 L/min. After the extraction, the average CO measured was 0.69 L/min, representing a 116% increase (Figure 4). However, there were notable variations in the MAP. In Case 1, the average pre-procedural MAP was 78 mmHg. This increased to an average of 78.93 mmHg post-procedure, indicating a 103% increase. On the contrary, in Case 2, the average MAP decreased by 5%, pre- and post-procedure, from 83.33 mmHg to 78.87 mmHg (Figure 5).

## 4. Discussion

Sufficient oxygen delivery to tissues is important and essential for survival. To achieve this, cardiovascular monitoring and management before and after surgery are pivotal for patient care. This is especially relevant for patients with HW disease, because blood flow is hindered by HW infection. In patients with severe HW disease, the surgical removal of the adult HWs is necessary to restore the cardiovascular circulation. However, research on the CO in HW-infected dogs has been limited. The most recent study was published in 1991 [11], which examined cardiopulmonary function values before and after HW extraction in dogs with caval syndrome.

In veterinary medicine, research on various HW removal devices has been conducted. However, it has several limitations, including the potential to induce cardiac damage [12], unsuitability for small animals due to the development being tailored for human applications, and a low retrieval rate per extraction, resulting in a prolonged procedure [13,14,15]. The critical factor in selecting the HW removal device for this case was to minimize interference with factors affecting CO changes by swiftly removing a significant number of HWs within a short timeframe. The HW basket device (S & G Biotech Inc., Seoul, Republic of Korea) was designed for veterinary use and developed in 2021 for the primary purpose of HW extraction. It features a small diameter suitable for small dogs and nitinol wire, which minimizes the risk of tissue damage. Additionally, it allows for the easy manipulation and placement of the device in the desired locations within the right atrium, right ventricle, and main pulmonary artery due to its optimal angle. Furthermore, its basket-type design offers the advantage of extracting a substantial quantity of HWs in a single operation. These two key advantages make it particularly suitable for research aimed at assessing changes in cardiac output during HW removal.

Invasive methods used to measure CO can potentially impact the survival of patients with HW disease, thus their clinical application is challenging. Although various methods for measuring CO exist, ongoing research is actively exploring the accuracy of non-invasive methods [16], particularly in small animals [17,18,19]. ICG is a novel non-invasive method that has emerged recently for monitoring stroke volume (SV) and CO in human patients [20]. This technique measures the SV and CO during the cardiac cycle based on changes in the electrical conductivity of the thorax accompanying each heartbeat. Recent studies have demonstrated its effectiveness, especially in patients with technical difficulties or a need for non-invasive approaches, such as human neonatal and pediatric patients [21]. Studies investigating its reliability in veterinary patients and the feasibility of its clinical application are currently underway [22,23,24,25,26,27,28]. In patients with HW disease, the traditional invasive method for measuring CO using a pulmonary artery catheter is time-consuming, and can potentially interfere with the extraction process. Therefore, measuring the CO using ICG was selected for our two cases.

Blood pressure is affected by changes in CO and systemic vascular resistance (SVR) [29]. SVR is determined by the vascular anatomy, and vascular, tissue, and neurohumoral factors. Given that SVR can change depending on the patient’s condition, monitoring both the MAP and CO provides a more comprehensive understanding of the patient’s overall status. Blood pressure does not instantly respond to changes in CO due to the complexity of systemic vascular resistance. Even if the CO increases, the blood pressure can decrease due to the toxins inside the patient with HW disease [30]. Even with an increase in the CO, an immune response triggered by HW toxins can lead to hypotension. Notably, it is not possible to determine whether an immune reaction is induced by HW toxins based solely on hypotension during extraction. Alternatively, in cases where blood pressure drops due to cardiac injury or a decrease in blood volume during the surgical procedure, it is necessary to consider discontinuing the procedure or using medications or fluid therapy that act on the heart rather than immune-suppressing drugs. By simultaneously monitoring changes in the blood pressure and CO, clinicians can accurately assess the patient’s condition and respond according to the available evidence for specific situations (e.g., administering drugs to reduce the immune response, adjusting the fluid resuscitation, or proceeding with or discontinuing the procedure). Measuring CO can be beneficial for enhancing the success and survival rates of the extraction procedure.

In Case 1, changes in the CO directly led to an increase in the MAP, while in Case 2, changes in the CO did not affect the MAP. Possible reasons for this include an increase in the SVR, such as the release of HW toxins. Therefore, in Case 2, drugs were administered to reduce immune hypersensitivity. Furthermore, although the patient did not experience a significant drop in blood pressure, post-procedure hospitalization was extended by a day relative to Case 1. A previous study examined the CO in the right heart after HW extraction and reported that, in those that did not survive, the reflux persisted, and the CO did not increase [11]. Inferring from this, the increase in CO after the extraction procedure, and the ongoing survival in our two cases, our patients may be considered as having a good prognosis.

This report is limited, as we only reported on two cases. Increasing the patient sample size is a viable approach. Furthermore, conducting research using experimental animals to control variables such as age, cardiac conditions unrelated to HW, and the grade of HW disease could enable a more detailed observation of changes in CO. Additionally, further research on CO measurement using ICG in dogs infected with HW is also necessary. In the future, studies with larger sample sizes are warranted. The insights gained from our current two cases may contribute to the design of future studies as follows. First, studies can focus on the accurate prediction of patient monitoring plans based on the degree of recovery after HW extraction, which could minimize unnecessary resource utilization and costs while preventing premature monitoring discontinuation and potential mishaps. Second, clinically, while there are patients who come specifically for HW treatment, there are also cases where HW disease is detected as an incidental finding during pre-anesthetic evaluations for other surgical needs. If the HW infection is severe and affects anesthesia, extraction surgery is prioritized over the originally planned operation. To date, no studies are available to provide definitive guidance on the level of vital sign recovery needed for additional surgical interventions. Through further research, clinicians can evaluate patients and determine the appropriate starting point for treating pre-existing conditions that are co-present with HW disease. Third, the author previously had experience with a case in which the entire procedure was stopped due to instability in the patient’s vital signs during HW extraction surgery. Further exploration of this approach could potentially facilitate advanced anesthetic monitoring during procedures, enable a detailed identification of the underlying causes of aberrant situations, and offer the potential for problem resolution without procedure discontinuation. A method that enables a more successful resolution of problems during procedures could significantly enhance the success rate of HW extraction surgery. Finally, not just for HWs, this research can serve as a foundational study for diseases, such as gastric dilatation, volvulus, and cardiac tamponade, that impact CO and require immediate treatment for the restoration of CO, thus potentially providing benefits for the management of other diseases.

## 5. Conclusions

HW disease is a condition that disrupts blood flow, leading to a decreased CO. ICG was used during interventional HW extraction for the effective monitoring of CO during the anesthesia stabilization phase, and during and after surgery in our two cases. Based on these findings, we recommend future research studies to investigate the use of ICG for CO monitoring in diseases that require abnormalities in CO to be corrected.

## Figures and Tables

**Figure 1 animals-13-03127-f001:**
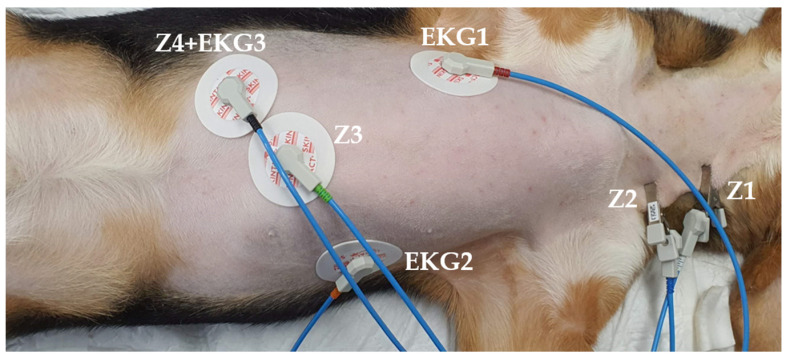
Electrode placement for the impedance cardiography measurement. The right side indicates the direction of the patient’s head.

**Figure 2 animals-13-03127-f002:**
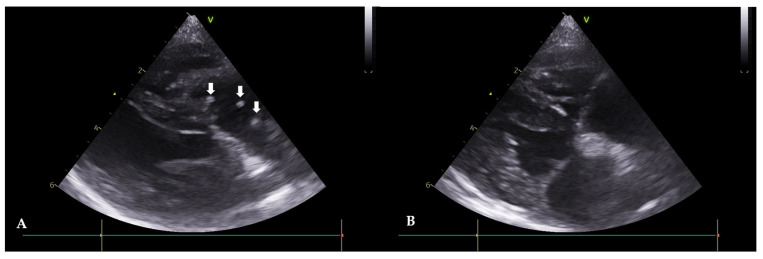
Echocardiograms of Case 1 before (**A**) and after (**B**) the heartworm (HW) removal (right parasternal short axis view). (**A**): Before the procedure, HWs are observed in the right atrium and ventricle (white arrows). (**B**): After the procedure, HWs are not observed.

**Figure 3 animals-13-03127-f003:**
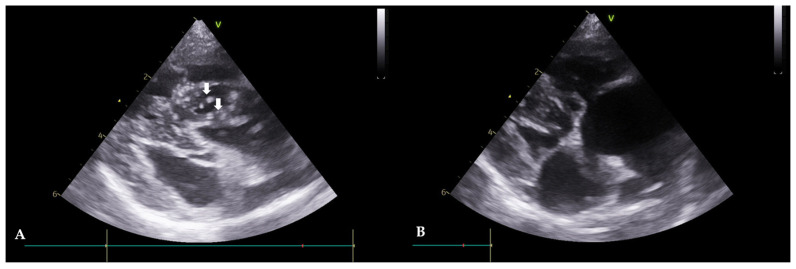
Echocardiograms of Case 2 before (**A**) and after (**B**) HW removal (right parasternal, short axis view). (**A**): Before the procedure, HWs are observed in the right atrium and ventricle (white arrows). (**B**): After the procedure, HWs are not observed.

**Figure 4 animals-13-03127-f004:**
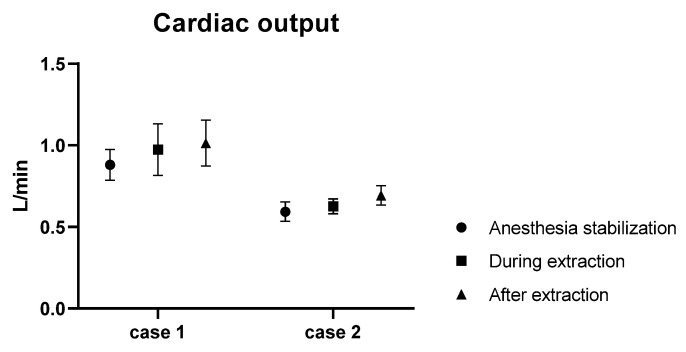
Graph showing the change in CO before, during, and after HW extraction.

**Figure 5 animals-13-03127-f005:**
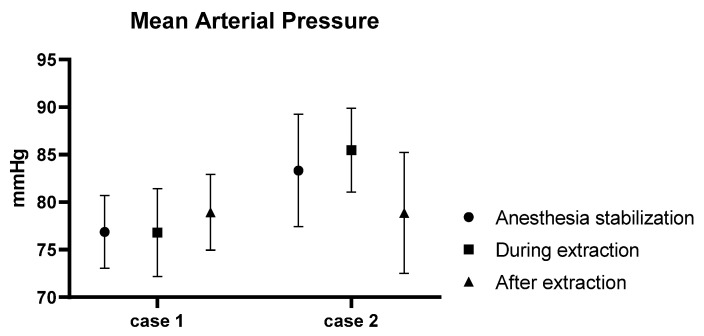
Graph showing the change in MAP before, during, and after HW extraction.

**Table 1 animals-13-03127-t001:** CO and MAP measurements before, during, and after the surgical extraction of HWs (n = 15, M ± SD).

Hemodynamic Event	Cardiac Output (L/min)		Mean Arterial Pressure (mmHg)	
	Case 1	Case 2	Case 1	Case 2
Anesthesia stabilization (before extraction) *	0.88 ± 0.09	0.59 ± 0.06	78.86 ± 3.68	83.33 ± 5.71
During the extraction process	0.97 ± 0.15	0.62 ± 0.04	76.8 ± 4.46	85.46 ± 4.25
After extraction	1.01 ± 0.13	0.69 ± 0.06	78.93 ± 3.86	78.87 ± 6.14
Rate of change * (%)	115%	116%	103%	95%

* Rate of change: the change in values from before extraction to after extraction was expressed as a percentage.

## Data Availability

The data presented in this study are available on request from the author.

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
