# Peer review of "Interventional Heartworm Extraction in Two Dogs: The Clinical Application of Impedance Cardiography"

_animals, 2023, doi:10.3390/ani13193127_

Round 1
Reviewer 1 Report
General comment: Authors could reduce using abbreviations and use them only when their use is absolutely necessary.

Minor English language editing required to improve the readability of the manuscript.
There are few formatting errors across the manuscript that need authors attention.
Author Response
Thank you for your review and contribution to my paper
I am pleased to inform you that, thanks to your guidance, the revisions have resulted in a marked qualitative improvement in the paper. Your thoughtful comments have sharpened the focus of the research and strengthened its contributions to the field. I genuinely appreciate your role in this process and the positive influence you've had on my work.

Reviewer 2 Report
The manuscript indicates timely and current work and has clinical interest.
However, I think it has limitations that have been indicated. Therefore I consider that it should be published
Author Response
Dear Reviewer
I hope you are doing well. I wanted to express my sincere appreciation for your thorough review and constructive feedback on the recent revisions I made to my paper, specifically regarding the content added from Line 246 onwards, which addresses the limitations of the study.
Your insights and suggestions have been invaluable in enhancing the completeness and clarity of the limitations section. Your thoughtful comments have enabled me to provide a more comprehensive analysis of the study's constraints, which I believe significantly strengthens the overall quality of the paper.
If you have any additional comments or suggestions, please feel free to share them. Your input remains highly valuable to me, and I am committed to making further improvements based on your expertise.
Once again, thank you for your time and commitment to the peer review process.
Warm regards

Reviewer 3 Report
This is a good paper about interventional Heartworm extraction. It contains enough background information and sufficient scientific proof. However, some information about cardiac function before and after extraction is missing, including peak E flow speed, pulmonary artery peak speed, and aortic artery peak speed. And the author is encouraged to illustrate more information about the interventional Heartworm extraction technique itself in supplementation, including tips and selection of heartworm baskets.
The quality of the English language can be improved to smooth the reading.
Author Response
Dear Reviewer
I wanted to extend my heartfelt gratitude for your diligent review of my paper, which has been immensely helpful in enhancing its overall quality.
I'm pleased to inform you that I have incorporated your valuable suggestions and made significant revisions to the manuscript. Specifically, in Case1, I have added relevant information about cardiac function, beginning from Line 109 through Line 117. In Case2, I have integrated the content you suggested, spanning from Line 142 to Line 148. Additionally, I have included detailed insights on the interventional Heartworm extraction technique in Lines 190-204.
Your insightful feedback has played a crucial role in ensuring the accuracy and comprehensiveness of these sections. I genuinely appreciate your expertise and dedication as a reviewer, which has undoubtedly contributed to the improvement of my work.
Should you have any further comments, concerns, or recommendations, please do not hesitate to share them. Your ongoing support and guidance are highly valued.
Once again, thank you for your time and commitment to the peer review process.
Warm regards,

Reviewer 4 Report
Manuscript animals-2626614
This is a description of two cases of canine heartworm infection that have been ttreated with surgical extraction of the worms under impediance cardiography surveillance.
line by line comments:
line 12: I suggest to replace non-intrusive by non-invasive
line 14 and throughout the manuscript: if the abbreviation HW for heartworm is introduced, it should then be used throughout the manuscript. Now it is used very inconsistently.
line 19: please always use "infection" and not "infestation"
paragraph 2.2: many sections are repeated.
line 95. delete sentence "Using a ...heartworm."
line 98: was only one microfilaria detected? If there were multiple, please use "microfilariae"
line 99 & line 100: enlarged pulmonary arteries are mentioned twice in the same sentence. Consider fusing into one statement.
line 100: please check if "reverse D heart sign, VHS 11.5, " is correct. It is writtendifferently in line 125.
line 102: delete "were detected". Move (Figure 2A) from line 108 to the end of this sentence.
lines 104 to 108: delete the whole text from "Immunological tests... liver enzymes."
line 114: I suggest to replace "the follow-up study" with "intervention"
Figures 2 & 3: red arrows are a very poor choice of colour for people printing the manuscript it black-and-white. I suggest to use white arrows.
line 123: delete "The dog... kit."
line 132: "The abdominal.. .heart failure." This is an interpretaion and not a result. Better put this into discussion.
line 142: I suggest to replace "the follow-up study" with "intervention"
line 153: "immiticide" should be the active substance, i.e. melarsomine.
line 163 and Figure 4: is it ml/min or L/min
lines 227ff: delete "ICG, ...surgery." and move the remaining paragraph to line 198. Make sure you use "infection" and not "infestation"
Author Response
Dear Reviewer
Your dedication to improving the quality of the paper is deeply appreciated, and your insights have undoubtedly contributed to its overall excellence. I am thankful for your commitment to the peer review process and your willingness to help me refine my work.
Once again, thank you for your time, expertise, and the valuable feedback you provided. Your contributions have been instrumental in strengthening the paper.
Warm regards,
Line-by-line comments and answer
Line 12: I suggest replacing "non-intrusive" with "non-invasive."
I revised it according to your suggestion.
Line 14 and throughout the manuscript: If the abbreviation "HW" for heartworm is introduced, it should be used consistently throughout the manuscript. Currently, it is used inconsistently.
I apologize for not adhering to the guidelines. According to MDPI guidelines, abbreviations should be handled separately in the abstract and main text, as recommended by MDPI. We have followed this approach in our submission.
For reference, MDPI guidelines specify that abbreviations in the abstract, main text, and figure/table/scheme captions should be treated separately. This means you must define an abbreviation the first time you use it in each of these sections, potentially requiring you to define the same abbreviation three separate times. This practice is due to the fact that these sections are often presented independently. For example, indexing services typically display only the abstract, and figures can be browsed without the main text on the journal's website.
Regarding the use of plural nouns like 'Heartworms,' while MDPI guidelines do not provide specific instructions, we have followed the APA Style Guidelines, abbreviating 'Heartworms' to 'HWs' without additional explanation. We appreciate your understanding.
Reference:
https://www.mdpi.com/authors/layout#_bookmark14
https://academicguides.waldenu.edu/formandstyle/apa/more/abbreviations
Line 19: Please always use "infection" and not "infestation."
I revised it according to your suggestion.
Paragraph 2.2: Many sections are repeated.
I revised it according to your suggestion.
Line 95: Delete the sentence "Using a ...heartworm."
I revised it according to your suggestion.
Line 98: Were only one microfilaria detected? If there were multiple, please use "microfilariae."
I revised it according to your suggestion.
Line 99 and line 100: Enlarged pulmonary arteries are mentioned twice in the same sentence. Consider combining them into one statement.
I revised it according to your suggestion.
Line 100: Please check if "reverse D heart sign, VHS 11.5," is correct. It is written differently in line 125.
I revised it according to your suggestion.
I have revised the repetitive sentence in 2.2. Case 1. The Vertebral Heart Size (VHS) for Case 1 is 11.5, while in 2.3. Case 2, the VHS is 10.4.
Line 102: Delete "were detected." Move "(Figure 2A)" from line 108 to the end of this sentence.
I revised it according to your suggestion.
Lines 104 to 108: Delete the entire text from "Immunological tests... liver enzymes."
I revised it according to your suggestion.
Line 114: I suggest replacing "the follow-up study" with "intervention."
I revised it according to your suggestion.
Figures 2 & 3: Red arrows are a poor choice of color for people printing the manuscript in black-and-white. I suggest using white arrows.
I revised it according to your suggestion.
Line 123: Delete "The dog... kit."
I revised it according to your suggestion.
Line 132: "The abdominal... heart failure." This is an interpretation and not a result. It would be better to put this into the discussion.
I revised it according to your suggestion.
Line 142: I suggest replacing "the follow-up study" with "intervention."
I revised it according to your suggestion.
Line 153: "Immiticide" should refer to the active substance, i.e., melarsomine.
I revised it according to your suggestion.
Line 163 and Figure 4: Specify whether it is ml/min or L/min.
I revised it according to your suggestion.
Lines 227ff: Delete "ICG, ...surgery." and move the remaining paragraph to line 198. Make sure to use "infection" and not "infestation."
I greatly appreciate your understanding of my intention and the appropriate modifications you provided, which have improved the overall quality. Thank you very much for this.
